# Reliable Prediction of Caco-2 Permeability by Supervised Recursive Machine Learning Approaches

**DOI:** 10.3390/pharmaceutics14101998

**Published:** 2022-09-21

**Authors:** Gabriela Falcón-Cano, Christophe Molina, Miguel Ángel Cabrera-Pérez

**Affiliations:** 1Unidad de Modelación y Experimentación Biofarmacéutica, Centro de Bioactivos Químicos, Universidad Central “Marta Abreu” de las Villas, Santa Clara 54830, Cuba; 2PIKAÏROS, S.A., 31650 Saint-Orens-de-Gameville, France; 3Departamento de Ciencias Farmacéuticas, Facultad de Ciencias, Universidad Católica del Norte, Angamos, Antofagasta 0610, Chile

**Keywords:** Caco-2, regression, quantitative structure–property relationship (QSPR), KNIME, permeability

## Abstract

The heterogeneity of the Caco-2 cell line and differences in experimental protocols for permeability assessment using this cell-based method have resulted in the high variability of Caco-2 permeability measurements. These problems have limited the generation of large datasets to develop accurate and applicable regression models. This study presents a QSPR approach developed on the KNIME analytical platform and based on a structurally diverse dataset of over 4900 molecules. Interpretable models were obtained using random forest supervised recursive algorithms for data cleaning and feature selection. The development of a conditional consensus model based on regional and global regression random forest produced models with RMSE values between 0.43–0.51 for all validation sets. The potential applicability of the model as a surrogate for the in vitro Caco-2 assay was demonstrated through blind prediction of 32 drugs recommended by the International Council for the Harmonization of Technical Requirements for Pharmaceuticals (ICH) for validation of in vitro permeability methods. The model was validated for the preliminary estimation of the BCS/BDDCS class. The KNIME workflow developed to automate new drug prediction is freely available. The results suggest that this automated prediction platform is a reliable tool for identifying the most promising compounds with high intestinal permeability during the early stages of drug discovery.

## 1. Introduction

The majority of drug discovery programs are focused on the development of new bioactive molecules by the oral route [1]. Two of the most relevant considerations for the successful development of new oral drug candidates are adequate intestinal absorption and oral bioavailability values [2]. Considering that experimental in vivo determination of intestinal absorption and oral bioavailability parameters is not feasible until the late stages of preclinical development, in vitro cell cultures have long been used as a surrogate alternative [3].

Among the cell cultures developed for intestinal permeability screening, the Caco-2 cell line is one of the most widely accepted biological techniques [4]. This cell line comes from human colorectal adenocarcinoma cells and has been widely used as an in vitro cell culture model of the human intestinal mucosa. Its usefulness for assessing human intestinal permeability has been attributed to the spontaneous differentiation into columnar enterocytes and the similarities in morphology (e.g., polarity, tight junctions, brush borders) and function (e.g., various transport mechanisms) to human enterocytes [5]. The absence of mucus, the variable expression of metabolic enzymes, and the differences in paracellular porosity have been pointed out as major limitations for the direct extrapolation of Caco-2 cell line permeability to fraction absorbed. In addition, the long culture period (21–24 days) and the high experimental variability have been identified as drawbacks of the in vitro method [6]. However, the Caco-2 cell assay is considered by the industry as the “gold standard” for in vitro prediction of intestinal drug permeability and absorption [7]. In addition, the in vitro Caco-2 assay has been recommended for the development of provisional BCS (Biopharmaceutics Classification System) and BDDCS (Biopharmaceutics Drug Disposition Classification System) [8].

Currently, given the large increase in new chemical entities generated at the early stage of drug discovery, it is quite challenging to perform high-throughput drug screening. Thus, providing a fast, simple, and cost-effective method as a substitute for the in vitro Caco-2 system, the development of in silico Caco-2 permeability models with high-throughput capacity for early identification of problematic drugs is essential [9]. However, the difficulties in developing a robust in silico tool to predict intestinal permeability remain, yet without satisfactory solutions. Indeed, Caco-2 permeability is a dramatically puzzling process that can take place through numerous nonlinear pathways (influx and efflux transporters) [10]. The paucity of high-quality Caco-2 permeability data has limited the development of accurate models with a comprehensive applicability domain. It is very common to observe variations during in vitro permeability protocols among different research groups because the cultured cells may vary depending on culture conditions, the number of passages, age of monolayer, seeding density, differentiation stage, and the transport buffer used [11,12]. Recently, Lee et al. found substantial differences for absolute apparent permeability coefficients (*P_app_*) of compounds between datasets from various laboratories with high normalized RMSE values in the range of 0.46 to 0.58 [13]. In addition, involuntary misprints due to erroneous transformations of values or units and the poor information related to the assay conditions have caused a decline in the quality of the data. Considering the above factors, the collection of Caco-2 datasets for modeling purposes needs a good balance between quality (experimental consistency) and size. If the experimental variability in the training dataset is high, building a reliable computational quantitative structure–property relationship (QSPR) model is likely to be difficult.

Numerous classification and QSPR models have been developed to predict Caco-2 permeability based on a variety of physicochemical and physiological descriptors [14]. Even when regression models have shown good performance on small datasets (<500) [15,16,17], most published work refers to classification algorithms that have been able to classify Caco-2 permeability with acceptable statistics according to the cut-off value used to transform numerical values into categorical classes. The main drawback of these classification models is that no consensus has been reached on the best cut-off for modeling purposes and from the perspective of the practical application of the model [14]. While there are many studies that predict Caco-2 permeability, there are few that propose a pathway for making predictions of new molecular entities. This issue has limited the applicability of published models. Considering the importance of a numerical value for decision-making, more robust and useful Caco-2 regression models are still needed, capable of covering a broad spectrum of chemical diversity and a wide range of magnitude. It is also required to make available automatic systems that can quickly and easily predict new molecules, to allow the development of virtual screening procedures and to implement the FAIR principles to make data findable, accessible, interoperable, and reusable [18,19,20]. Here, we present an automated platform for predicting permeability in the Caco-2 cell line. This work shows the potential of recursive machine learning models to perform variable and data selection with the aim to obtain interpretable and reliable regression models. A new method for developing a consensus model was implemented too. The model was developed on a curated dataset of more than 4900 molecules and was validated for the early estimation of permeability rate for BDDCS and preliminary evaluation of absorbed fraction. This platform provides a free tool for virtual screening of Caco-2 permeability in large compound libraries. The early evaluation of this property during the drug design and discovery stages should facilitate decision-making, minimize the number of experiments, and promote a rational selection of potential orally active drugs.

## 2. Materials and Methods

### 2.1. Computational Tool

In this study, KNIME Analytics Platform version 4.4.2 [21] and its free community extensions were used to develop an automated workflow for the transformation, analysis, modeling, and visualization of Caco-2 permeability data.

### 2.2. Permeability Data Collection

Experimental values of Caco-2 permeability were collected from three publicly available datasets [22,23,24]. Three criteria were followed for this selection: to have more than 1000 instances, to have been used for regression tasks, and to have been published within the last 5 years. The first dataset (Set A) was published by Wang et al. in 2016 [22]. They used a relatively large dataset consisting of 1272 compounds to develop several models for the prediction of Caco-2 permeability using different machine learning methods in combination with a genetic algorithm for variable selection. The second dataset of 1827 compounds (Set B) was collected by Wang and Cheng [23] to develop several QSPR models based on neural networks and other machine learning approaches. The third dataset of 4464 compounds (Set C) was reported by Wang et al. in 2020 to model the Caco-2 permeability using neural networks approaches [24]. All permeability measurements were converted to cm/s × 10^−6^ and were transformed to a base 10 logarithmic scale to form a Caco-2 permeability dataset for modeling. Missing entries for permeability values were removed. To minimize the uncertainty and evaluate the experimental variability, the mean value and the standard deviation (STD) were calculated for the repeated entries. Considering that the validation dataset should have the highest possible quality, the MERGED data were split according to the standard deviation computed from molecules with multiple measurements. Samples with standard deviation known (STD ≠ 0) and equal to or less than 0.5 formed the reliable validation set. The rest of the molecules were tagged as UNCLEAN and used as part of the training set. To validate the model performance, an additional list of 100 commercial drugs was included as the external set [25].

### 2.3. Chemical and Experimental Data Curation

Recommended good practices for data curation were followed to ensure a correct molecular representation and eliminate unreliable samples [26]. The curation workflow was split into three main steps: cleaning of chemical structures, standardization of the molecular representation, and treatment of duplicates. The final goal was to obtain a curated set made of unique molecules, ensuring the correctness of chemical representations before duplicate filtering and considering the variability of the target property during duplicate analysis (see Figure A1 in the Appendix B).

### 2.4. Physicochemical and Structural Descriptors

The calculation scheme was based only on the 2D structure; therefore, stereoisomers of the same molecule were considered duplicates and removed in the previous steps. Physicochemical properties, MOE-type and Kappa descriptors, and Morgan fingerprints (1024 bits) were calculated using the “*RDKit Descriptor*” and “*RDKit Fingerprint*” nodes available in the RDKit plugin of KNIME [27].

### 2.5. Variable Selection

A variant of the recursive variable selection algorithm published in a previous study was followed to simplify the model complexity, determine the most relevant variables, and minimize the number of correlated and uninformative features [28]. Firstly, a 10% cut-off for missing values was fixed for each molecular feature. Descriptors with constant values were excluded using a low variance cut-off of 0.1. Then, a random forest feature selection based on variable permutation and a correlation analysis was performed. Figure A2 in the Appendix B shows a schematic description of this procedure. Duplicates of every numerical descriptor (X_j_) were shuffled before training a regression random forest (RRF) model with shuffled (X_j shuffled_) and non-shuffled original variables (X_j_). Individual regression trees were extracted from the model and the number of occurrences of each shuffled and non-shuffled variable was determined by computing the total number of nodes that used X_j_ for splitting. Until this point, variables were kept if the ratio between the number of occurrences of the original variable and the number of occurrences of the counterpart shuffled variable was twice or more. The number of variables was recursively reduced by initially computing the Pearson correlation coefficient between them. If this linear correlation between two variables exceeded the threshold established (Pearson correlation coefficient ≥ 0.85), only the variable with the highest number of occurrences was kept to avoid the inclusion of redundant information in the model. Among correlated variables, continuous variables were privileged against discrete variables since they showed higher *r*^2^ and lower RMSE statistics in this study.

### 2.6. Data Cleaning

A threshold of 0.5 standard deviation (STD) between experimental log *P_app_* values was set for splitting the modeling data into two sets: the reliable set (STD ≤ 0.5 AND STD > 0) and UNCLEAN set (unknown STD, STD = 0.0 or STD > 0.5). For the UNCLEAN set, a recursive clean-up approach was developed. The algorithm begins by randomly splitting the unreliable set into two sets of 50%. Two regression random forest (RRF) models were trained in parallel for each set. Alternatively, the other set was used as a test set. For each test set, the percentage difference between the experimental and predicted value was calculated as follows:(1)$$\mathrm{Recursive}\;\mathrm{Prediction}\;\mathrm{Error}\;\mathrm{Percentage}\;(\mathrm{RPE})=\frac{\left|\log P_{app}^{\;\;observed}-\log P_{app}^{\;\;calculated}\right|}{\log P_{app}^{\;range}}*100\%$$

If the percentage error computed was higher than 30% (RMSE > 1.5 log units) and the prediction variance of the random forest model was lower than 0.1, the molecules were finally classified as failed samples. Through a recursive procedure, initially started from the first random partition, molecules were classified as passed or failed samples and assigned to the cleaned set or definitely tagged as failed data, respectively. The procedure finished when no more failed samples were detected.

### 2.7. Modeling Algorithm

To provide an unbiased estimate of prediction accuracy, a 10-fold cross-validation procedure was followed for model tuning. The optimization included the choice of thresholds used for variable selection, data cleaning, and applicability domain approaches. The modeling algorithm consisted of a conditional consensus model (CCM) made of individual regression random forests. First, the cleaned data were used to train a global regression random forest model (*ntree* = 51). The training samples were then grouped according to four permeability ranges (PR)—“low (L)” log *P_app_* < 6; “low–moderate (L-M)” 6 ≤ log *P_app_* < 5; “moderate–high (M-H)” 5 ≤ log *P_app_* < 4.7; “high (H)” 4.7 ≤ log *P_app_* to build four regional random forest models (*ntree* = 51)). The selection of these cut-off points is based on the literature review [29]. Metoprolol permeability (20 × 10^−6^ cm/s; log *P_app_* = −4.7) was selected to separate molecules with high permeability from the rest. The cut-off point −5 log units (10 × 10^−6^ cm/s) was chosen as a reasonable midpoint to separate molecules with moderate–high permeability, whereas the log *P_app_* cut-off point, 6 log units (1 × 10^−6^ cm/s) separated low-permeability molecules. To predict a new molecule, a similarity search found the five nearest neighbors in the training set above a Tanimoto coefficient of 0.7. The new molecule was assigned to the corresponding regional models based on the class of its nearest neighbors (NN). In this way, molecule permeability can be estimated by a combination of values coming from one to all four regional models. Up to this point, if the molecule was predicted by more than one regional model, the eligible predictions were averaged. To minimize the boundary effect of setting a “hard” cut-off after dividing the data into four training regions, the new samples were predicted by the global random forest regression too.

### 2.8. Model Evaluation

The coefficient of determination (r^2^ _validation_), root mean squared error (RMSE _validation_), mean absolute error (MAE), and the percentage of molecules within the range of 0–0.5 (log units) of the absolute difference between experimental and predicted log *P_app_* (% 0.5 log) were used for model evaluation [30].

## 3. Results and Discussion

### 3.1. Permeability Data

Permeability data were collected from three public data sources, which have been used independently to build QSPR models in previous studies [22,23,24]. Rigorous chemical and experimental data curation steps were performed before modeling to ensure the development of reliable models (see Methods for chemical and experimental data curation). This issue is of particular interest in this study, due to the combination of different data sources. After the chemical data curation procedure, the three original datasets A, B, and C consisted of 1267, 1863, and 4462 molecules, respectively (see Figure 1a). Before the removal of duplicates, the grade of agreement between the datasets was evaluated by comparing the log *P_app_* values reported for the same molecules. Figure 1a shows the intra- and interset RMSE statistics based only on molecules with different values reported in the same or different data sources (results are shown in the RMSE matrix). This comparison provides us with an estimation of the expected lower limit of root mean squared error prediction (RMSEP), since experimental errors should be always lower than computational errors from in silico models. Set B showed the best degree of agreement with the others, with RMSE values below 0.35 for all cases. The most critical value reported corresponds to the RMSE analysis between duplicate pairs in set C (RMSE intraset = 0.49). Note that the RMSE values obtained from the comparison of the three main datasets versus the external set ranged from 0.23 to 0.52.

To analyze the predictive limits of the QSPR model on the whole dataset (excluding the external set), we added a random error to the target property by sampling from a Gaussian distribution of zero mean and standard deviation σ:(2)$$\log P_{app}^{\;\;\;\;\;noise}=\log P_{app}^{\;\;\;\;\;true}+N\left(0,\sigma \right)$$

Two levels of standard deviation were evaluated in correspondence with the mean and maximum values of RMSE reported in the matrix depicted in Figure 1a. After 10 replicates for each level, the coefficient of determination (r^2^) for the whole set ranged from 0.55 to 0.82. The same procedure was applied to the external set. For this set, the r^2^ values ranged from 0.42 to 0.84.

To reduce experimental uncertainty, only those molecules with the same chemical structure and standard deviation ≤0.5 were considered as duplicates. Manual inspection confirmed that the pairs corresponded to truly duplicate structures. Among 1836 molecules with more than one experimental value, only 9 pairs corresponded to stereoisomers. From this list, only one pair presented a significant difference in the log *P_app_* values with a standard deviation of 0.79 (Pravastatin). This molecule was excluded from the modeling dataset. The rest of the stereoisomers were in the 0.0–0.5 standard deviation range.

Finally, the analysis of 2D duplicates identified 5000 unique molecules. Before modeling, it was ensured that there was not any overlap between this set and the external set. This way, 87 molecules were eliminated from the modeling dataset, herein called MERGED data (4913 molecules).

### 3.2. Molecular Properties and Caco-2 MERGED Data

Figure 1c shows the distribution of the count of failures of the extended rule-of-five (Ro5) on the MERGED data. Following this rule of thumb, more than 60% of the MERGED data satisfied the extended Ro5 guidelines for oral bioavailability and the rest failed in at least one condition [31,32]. The ranking of molecular properties by increasing number of failures was as follows: number of hydrogen bond donors (HBD), calculated partition coefficient (slogP), number of hydrogen bond acceptors (HBA), number of rotatable bonds (RBN), and molecular weight (MW). Appendix A shows the frequency distribution histograms of all these properties on the MERGED dataset.

The analysis of the dataset showed a slight trend of increasing permeability when decreasing HBD, HBA, MW, and RBN. The opposite effect occurred with the slogP lipophilicity descriptor. Although the correlations of all these properties with the log *P_app_* values in our data were not very strong (Pearson correlation coefficient <0.4), their impact on the permeation rate has been widely described [10,25,33].

Lipophilicity measured as logP is a structural parameter encoding intermolecular forces (electrostatic and hydrophobic) and intramolecular interactions. This metric represents the ratio at the equilibrium of the concentration of two phases (oil and aqueous phases) [34]. High values are associated with more lipophilic compounds that are more capable to cross lipidic layers via a passive diffusion [35].

The impact of the hydrogen bonds’ acceptor/donor count on the interaction between the molecule and potential targets as the glycoprotein P has been widely recognized [36]. Indeed, HBA and HBD counts have been used to estimate P-gp substrate recognition [37]. Regarding passive diffusion, the functions of hydrogen bonding in solute–solvent interactions explain its influence during molecule permeation. Several studies have proposed that when the number of solute–solvent H-bonds increased, permeability decreased. This observation has lead to chemical transformations by N-methylation to increase drug permeability. In addition, other studies have demonstrated that the introduction of intramolecular hydrogen bonds has yielded to a permeability increase since it reduces the exposed HBD count [10,36].

A tendency of poor permeability for molecules with molecular weight higher than 500 g/mol (75% of the molecules in this group had Caco-2 permeability values below 10 × 10^−6^ cm/s cut-off) was observed. However, the opposite effect was not observed in the molecules with low molecular weight. This reinforces the idea suggested by other authors that the influence of molecular weight on oral absorption needs to be evaluated together with a flexibility metric as the number of rotatable bonds. The number of rotatable bonds is related to molecular size and flexibility. Both properties can affect passive diffusion, active transport, and P-gp interaction. Since molecular size measures the area occupied by the molecule in its 3D space, it is obvious to expect a considerable influence of this property on drug transport by paracellular and transcellular routes. Veber et al. studied the effect of molecular flexibility, measured as RBN, but in the context of molecular weight [31] This analysis was applied to our data and showed that the influence of the rotatable bond count on Caco-2 permeability depends on molecular weight. We split the dataset at 500 g/mol and then, we divided it into four RBN ranges: [0,5], (5,10], (10,15], and (15,62]. The choice of this molecular weight threshold follows the Lipinski threshold recommendation for drug-likeness. Molecular weight equal to or less than 500 g/mol and RBN ≤ 5 appears to be a better indicator of high permeability. More than 60% of the molecules in this group showed permeability values above 10 × 10^−6^ cm/s (cut-off associated with good oral absorption), in contrast to 14% of molecules in this RBN range but with a molecular weight above 500 g/mol. Similar behavior was observed in the following range of RBN: molecules with lower molecular weight are more likely to exhibit high permeation rates in the Caco-2 cell line when RBN is higher than 5 or equal to or lower than 10. In the last range of RBN, it was observed that, for high molecular weight molecules, the presence of a high number of rotatable bonds can enhance drug permeability since the free rotation of atoms around single bonds allows the molecule to adjust its conformation during permeation. These relationships are reasonable considering that different transport mechanisms govern drug permeability and are expressed in the Caco-2 cell line.

In addition to lipophilicity and molecular size, the impact of molecular polarity on passive diffusion has been described [32,38]. TPSA has been frequently used as a descriptor in Caco-2 permeability models to describe its effect on passive diffusion, and its relationship with the strong H-bond interactions between the Caco-2 cell line and drugs [39]. Molecules with high TPSA may exhibit poor in vivo permeability due to the influence of polarity on the interaction of the drug with glycoprotein P (P-gp) [25].

### 3.3. Model Development

Figure 2 provides a schematic description of model development. Considering the need to contextualize the model performance against the experimental variability, the reliable samples (*n* = 728) formed the first validation set. The UNCLEAN data (*n* = 4185) were used to select the minimum number of features capable of correlating chemical structure with permeability (see Methods for variable selection). The SMILES codes and the RDKit descriptors of each structure are available as Appendix A. Initially, more than 20 uninformative variables with variance lower than 0.1 were excluded, and no missing values were reported for any RDKit descriptor. The first part of the recursive selection algorithm consisted of retaining variables whose number of occurrences in the random forest model was twice the number of occurrences of the corresponding permuted variable. Until this point, 34 variables were pre-selected. The Pearson correlation matrix for all these variables was computed and the number of variables was recursively reduced by eliminating the correlated variables with a lower ratio of occurrences. An average of 15 shortlisted variables were selected using this algorithm.

Then, a supervised recursive clean-up algorithm was developed to discard unreliable samples and to form a cleaned training set (see Methods for recursive data cleaning). Once the failed samples were discarded (50 ± 10 molecules), the resulting cleaned data were used as the training set. Comparisons of the molecular descriptor space in the first two principal components indicate that the reliable validation set and cleaned (passed samples used as training set) occupied a similar space (see Appendix A). This fact reinforces the idea that the reliable validation set prediction could provide a realistic estimation of model performance.

Five regression random forest (RRF) models were trained using different training spaces to build a conditional consensus model (CCM). The first model consisted of a global RRF model trained on all cleaned samples and thus covering all the PR. Then, cleaned samples were clustered according to four classes of PR: “low (L)”, “low–moderate (L-M)”, “moderate–high (M-H)”, and “high (H)”. One regional RRF was trained on each cluster. We define a regional RRF as an RRF only trained within one of the previously defined permeability clusters. Thus, a regional RRF is not a local model in terms of molecular description but a local model in terms of the property to be predicted. A new molecule was predicted first by the global RRF and then by the corresponding regional models if the Tanimoto coefficient (similarity metric) to its nearest neighbors was equal to or greater than 0.7. The algorithm employs Tanimoto distance to find the five training instances closest to the new molecule.

The percentage improvement of the RMSE-CV for the conditional consensus model (CCM) over the global random forest model ranged from 6% to 22%, showing that the use of regional models can reduce the prediction error of test samples with high similarity to the training samples. Validation metrics for the cross-validation, reliable, and external sets are depicted in Table 1. The results are reported as the mean with standard deviation.

Different statistical results were computed after data and attribute randomization using different seeds in the setup of the regression random forest during 11 independent runs. Regression plots for the reliable set and external set are shown in Appendix A.

After completing the 11 training runs of this model, a complementary variable selection was performed on the previously selected variables. A recursive feature elimination was performed by starting with the training of an RRF model with all the features previously selected and successively removing the features in decreasing order of importance until the degradation of the model in terms of the mean absolute error of the out-of-bag samples was observed. During this recursive process, it was observed that discrete variables were less performant than their equivalent correlated continuous variables, if any. Based on this observation, the most correlated continuous variables with the higher number of occurrences were always taken rather than their correlated discrete counterpart variables. Thus, the algorithm favors the selection of uncorrelated variables with a high number of occurrences and prefers continuous variables over discrete variables. From the initial list of the most important variables, the features slogP (octanol-water partition coefficient), TPSA (topological polar surface area), SMR (molecular refractivity), Halkier Alpha (Hall–Kier alpha value), and Kappa 3 were selected. The influence of the descriptors slogP and TPSA have been previously described. The SMR descriptor, as a metric of the molar refractivity, is related to the volume of the molecule and the London dispersive forces. Both parameters can affect the size, the polarizability [40], and the passive diffusion through the lipid barrier. The Halkier Alpha value quantifies the molecular shape and encodes the effects of the covalent radius and hybridization stage [41]. This molecular descriptor and its related approaches have been identified in previous Caco-2 computational studies [16,29]. Kappa 3 (molecular shape descriptor) appears as a surrogate of the discrete variable number of rotatable bonds (RBN) because it is the most correlated continuous variable with the higher number of occurrences in the random forest model trained for variable selection. Regarding the use of RBN as a metric of molecule flexibility, it has been described that the discrete nature of this variable and the exclusion of cyclic moieties from the count can be problematic [42]. We note that using Kappa indices as descriptors of molecular flexibility instead of the count of rotatable bonds improved the permeability prediction. Appendix A show the importance of these variables measured in terms of occurrences in the RRF model and the correlation matrix between all of them, respectively. These five variables were used to construct a new lower-dimensional model. This time, RMSE values ranged from 0.46–0.53, and r^2^ values were 0.5 (0.01), 0.56 (0.01), and 0.56 (0.02) for the cross-validation, reliable, and external sets, respectively. These results highlight the consistency of the model and demonstrate that the recursive variable selection algorithm is useful in identifying the minimum number of variables good enough to predict the target property.

The predictive ability of a QSPR model must be evaluated according to the experimental variability of the in vitro test. Note that the validation metric of the reliable set presents the best coefficient of determination and the lowest root mean squared error. However, if we select a 15% random set of molecules (size of the reliable set) from the whole MERGED Caco-2 dataset, the statistics are very close to the results of the cross-validation set. Although r^2^ is a popular metric for regression, a measure of dispersion such as the RMSE is a more useful indicator of model accuracy than r^2^ [30]. Indeed, r^2^ compares the variance of the residuals to the observations themselves, whereas the RMSE is not influenced by the size or distribution of the data and provides a quantitative measure of the standard deviation of the residuals. In terms of root mean squared error, the predictive ability of the model is acceptable according to the experimental uncertainty of the data sources, demonstrated by the RMSE calculated among the purely experimental values during the data exploration. Following the rule that the RMSE of the test set should be less than 10% of the range of the target property [30], and considering that more than 69% of the molecules of all validation sets fall within the 0.5 log range of the prediction error, the model is suitable for predicting the log *P_app_*.

Table 2 compares the results achieved by the conditional consensus model (CCM) with previous studies. To avoid the impact of data size on the comparison of models, this table only shows QSPR studies based on more than 1000 samples. Regarding data size, our model is ranked second only surpassed by the Sherer et al. study, which included more than 15,000 samples (non-public data) [43]. It is well known that final model quality is strongly dependent on training set quality, diversity, and size. Consequently, it is usually at risk to compare numerical results based on datasets of different compositions and sizes. In terms of RMSE, our model achieved comparable results to previous models, considering the data size, the range of permeability, and the wide experimental variability demonstrated in the data exploration.

The functionality of the model can also be described in terms of classification metrics. Based on the cut-off point associated with complete absorption (*P_app_* = 10 × 10^−6^ cm/s), the results of the model predictions showed high accuracy for all validation sets (accuracy: 0.77–0.82; sensitivity: 0.61–0.77; specificity: 0.87–0.91) These results are comparable to those achieved in other studies [22,29,45,46]. Although the cut-off point for establishing an acceptable rank–order relationship between permeability values in Caco-2 and the human absorption of drugs is still under debate, the selected *P_app_* value represents a reasonable midpoint between very stringent or very soft cut-off values (e.g., 20 × 10^−6^ cm/s and 2 × 10^−6^ cm/s) [46].

### 3.4. Applicability Domain

The reliability of the model predictions was evaluated by the combination of two approaches. The first approach consisted of estimating the prediction error of new samples through the five nearest neighbors (5-NN) algorithm based on the Euclidean distance. The estimated prediction error of the new samples was computed as follows: $\mathrm{Estimated}\;\mathrm{Prediction}\;\mathrm{Error}\;\left(\%\right)=\left(\frac{\sum_{1}^{5}RPE_{NN}}{5}\right)$, where *RPE_NN_* is the recursive prediction error percentage computed for the nearest neighbors during the recursive data cleaning algorithm. The second approach consisted of comparing the prediction variance computed from random forest individual predictions with a predefined optimized threshold of 0.3. New samples with highly variable predictions (prediction variance >0.3) and with an estimated prediction error greater than 10% (RMSE > 0.5) were considered low confidence. Figure 3 shows the comparison between the error metrics for the high and low confidence levels. This method proved its suitability for identifying unreliable samples. The percent of decreasing of the RMSE values for the high confidence level ranged between 25% and 29% for the validation sets.

### 3.5. Model Validation with Reference Drugs and Potential Application to BCS/BDDCS

Recently, according to the Food and Drug Administration (FDA) and the International Council for the Harmonization of Technical Requirements for Pharmaceuticals (ICH) guidelines [47], the suitability of a cell-based model to estimate drug permeability for BCS-based biowaiver must be experimentally confirmed based on reference drugs within a wide range of oral absorption and different transport mechanisms [47,48]. In this sense, an additional external set made of 32 reference drugs recommended by the ICH was assembled to evaluate the model performance.

Since the ICH guideline does not report the quantitative permeability value for these molecules, another 10 sources were consulted [7,25,35,49,50,51,52,53,54,55]. For molecules substrates of efflux mechanisms, it was ensured to take the value corresponding to the assay without the presence of any efflux inhibitor. In relation to the experimental conditions, assays with extreme values of stirring rate and pH of transport buffer were not considered. The standard deviation distribution of the experimental values collected from the literature for the 32 reference drugs are depicted in Figure 4d. Note that the major number of molecules is distributed in the range 0.2–0.6 standard deviation. These results are consistent with the high variability reported in other studies. The median value of all experimental measurements was adopted as the final value to evaluate the model performance. To implement this task, the model was retrained on the whole MERGED dataset and only those molecules belonging to the new external set were removed to ensure a blind prediction. Model predictions for these drugs are shown in Table 3.

The regression and the residual plots are depicted in Figure 4a,b, respectively. In correspondence with the 10 × 10^−6^ cm/s (−5 log units) cut-off for high permeability, the model allows us to correctly identify 100% of the molecules with moderate–low permeability and 89% of the molecules with high permeability. In terms of the prediction of the quantitative value, the model was able to predict the new molecules with an r^2^ mean value of 0.55 ± 0.07, and around 72% ± 5 of the molecules were predicted within the range of 0.5 of log units. To estimate the ideal predictive limits for this set, the mean value of each molecule computed from multiple measurements was compared to every single measurement value reported in the literature. This correlation analysis showed an RMSE of 0.4 and r^2^ of 0.7 (Figure 4c). Theoretically, r^2^ = 0.7 and not r^2^ = 1 would be the upper predictive limit of any computational model for this set; thus, the r^2^ reached by the CCM method is completely reasonable in the context of the experimental variability. Residual values higher than 0.5 were reported for molecules with moderate–low permeability (see Figure 4b). There are many factors that affect model performance in this range. First, only 35% of the data covers the experimental range of −5.5 to −8.5. In addition, a slight tendency to increase experimental variability was observed for low permeability molecules. Most of the drugs with larger discrepancies are substrates of transporter proteins and this phenomenon cannot be completely covered by the physicochemical properties used by the model.

The relationship of the apparent permeability coefficient with the fraction absorbed and the extent of metabolism are displayed in Figure 5, respectively. The good correlation between the apparent permeability calculated (expressed as *P_app_* × 10^−6^ cm/s) with the fraction absorbed (Fa) (*Spearman’s rank* correlation coefficient = 0.77 ± 0.01) showed that the current in silico data (*P_app_* estimations) could be used to give an initial evaluation of in vivo absorption, independently of the mechanisms of transport or efflux that could govern human permeability. Similar performance was observed for the relationship between the in silico log *P_app_* data and the extent of metabolism (*Spearman’s rank* correlation coefficient = 0.8 ± 0.01).

In the context of drug discovery, early prediction of the BCS or BDDCS class for new molecular entities plays a critical role [8]. Provisional classification in both systems can provide an early estimation of in vivo drug performance, transport mechanisms, and drug disposition [56]. While the BCS classification system uses drug permeability to predict drug absorption, the BDDCS proposes the use of permeability rate (as a measure of the extent of metabolism) [57]. Several studies have suggested the use of in vitro permeability in cell lines such as PAMPA, MDCK, and Caco-2 for estimating drug permeability or the extent of metabolism [58,59,60]. Other approaches have relied on the use of molecular descriptors of polarizability and lipophilicity [61,62,63], while other contributions are based on the use of machine learning models for predicting the permeability/extent of metabolism from the molecular structure for further application on BCS/BDDCS [64,65]. In this sense, the potential applicability of conditional consensus model predictions to Provisionals BCS/BDDCS was evaluated. It is important to note that this provisional classification is limited to drug permeability/the extent of metabolism.

For BCS provisional classification, the cut-off for in silico permeability was fixed to 10 × 10^−6^ cm/s, as used for other studies [66]. For the ICH list, BCS classifications were compiled from the literature [8]. Among the 21 drugs on the ICH list with known and unambiguous classification, 18 were correctly predicted with an accuracy of 86%.

For BDDCS, the optimal threshold to define the boundary between high and low permeability was selected using a dataset of 679 drugs as reported by Brocattelli and coworkers [65]. The BDDCS classifications were updated according to the new classifications provided in the latest article by Bocci et al. [8]. Permeability predictions were calculated for this set, previously ensuring that any overlap between this set and our MERGED set was removed. From the initial BDDCS dataset, 70% was randomly chosen as the training set. The remainder was used as the test set. Receiver operating characteristic (ROC) analysis on the BDDCS training set suggested that an in silico log *P_app_* permeability cut-off of 6.3 × 10^−6^ cm/s can be used to discriminate between classes 1–2 (high permeability rate) and 3–4 (low permeability rate). By setting this threshold, the in silico permeability data were able to correctly identify 80% of test set drugs in class 1–2 drugs and 70% in class 3–4. If low confidence samples were removed, the accuracy reached 81% for 191 molecules (94% of the test set). In the case of the ICH list, 26 drugs out of a total of 32 were correctly classified in classes 1–2 or 3–4, for an accuracy of 81%.

To compare our results with previous in silico BCS/BDDCS, the ICH list was used as the reference set. Only purely in silico studies in which model predictions were publicly available were considered for this comparison (see Table 4).

Table 5 reviews the main in silico contributions to the provisional BCS/BDDCS. Three studies use the partition coefficient as a descriptor to predict the permeability class or the degree of metabolism. However, it has been described that lipophilicity alone is not a good indicator to characterize drugs with carrier-mediated active transport. Furthermore, the relationship between lipophilicity and the degree of metabolism deteriorates for drugs that are eliminated via the biliary route. Both facts lead to incorporating other characteristics to achieve a better concordance. From Table 5, it can be observed that models that use polarizability metrics and logD reach better results in global terms. However, the comparison of the models is limited by the different test sizes, chemical space representations, and the recent updates to the BCS/BDDCS classifications. In this study, in silico log *P_app_* values allowed for the correct identification of 75% of drugs with active transport (see Table 3), which proved to be a better descriptor to discriminate between classes 1–2 or 3–4. We consider that the current approach shows acceptable performance on both test sets and provides a good and fast method for preliminary BCS/BDDSC classifications, highlighting that the predictions are fully automated in a KNIME workflow able to classify new molecular entities.

### 3.6. Automated Platform for In Silico Caco-2 Permeability Prediction, Concluding Remarks

The KNIME workflow can be downloaded free of charge from https://pikairos.eu/download/Caco-2-permeability-prediction (accessed on 16 September 2022). The user can directly predict the Caco-2 permeability of the new molecules from SMILES codes simply by following the KNIME installation instructions and the workflow usage guidelines. The workflow was designed to provide three outputs for permeability analysis: (1) the numerical value of the apparent permeability expressed as *P_app_* × 10^−6^ cm/s; (2) the permeability class according to the 10 × 10^−6^ cm/s cut-off; (3) the preliminary BDDCS/BCS classification into classes 1–2 or 3–4; and (4) the confidence level of the estimators. Thus, the user can prioritize the molecules with high permeability and high confidence levels. The use of a large and diverse dataset and the combination of regional and global regression models allowed us to cover a broad spectrum of chemical diversity and provide a comprehensive applicability domain. The mathematical treatment to take relevant information from this data is novel and the results showed a good correlation in the context of experimental variability. The usefulness of this system lies in its simplicity, since no experimental properties are used as input features, and it is only based on easily calculated 0–2D physicochemical and structural descriptors.

## 4. Conclusions

A novel methodology was applied to model Caco-2 permeability on a reasonably large dataset. This approach combines recursive algorithms for variable selection and data cleaning with a conditional consensus model (CCM) made of regional and global random forest models. We presented one of the most extensive and fully public datasets used for Caco-2 modeling. Even when experimental conditions (pH, stirring rate, cell passage number, temperature, and cell culture conditions) were not available, the methodology was able to achieve good performance on a very heterogeneous dataset. The validation statistics of the model improved with the quality of validation sets in terms of the standard deviation of molecules with multiple measurements. The practical use of the model was confirmed with the blind prediction of a reliable validation set and two commercially available drug datasets, reaching RMSE values between 0.43 and 0.51 log units. The results obtained are consistent with the experimental variability of the modeling dataset. The methodology is based on a reduced number of basic molecular properties or 0–2D descriptors and is easily reproducible and applicable. Considerable curation efforts were made before modeling to avoid overestimation of the QSPR models and to ensure the validity of the structure–permeability relationship model. The model will be updated with new data to incorporate more chemical information and provide an extensive applicability domain. From a practical perspective of the Caco-2 regression model, the workflow for prediction is publicly available and includes as outputs the numeric value of apparent permeability on Caco-2 and the corresponding confidence level of the estimators to facilitate a Caco-2 prediction tool for the medicinal chemistry community. In addition, the model proves to be useful for discriminating between classes 1–2 and 3–4 of the BCS and BDDCS. Finally, we consider that the present in silico model is a reliable tool to identify the most promising compounds with high intestinal permeability during the early stages of drug discovery and could be applied in the development of further provisional BCS or BDDCS classification systems. The performance of these new QSPR models is also expected to improve, as more and higher-quality data with corresponding experimental conditions are released in the future. The construction of QSPR models based on well-characterized Caco-2 data will be a step forward in the prediction of this task since individual models could be built for each range of experimental conditions without combining measurements from different laboratories.

## Figures and Tables

**Figure 1 pharmaceutics-14-01998-f001:**
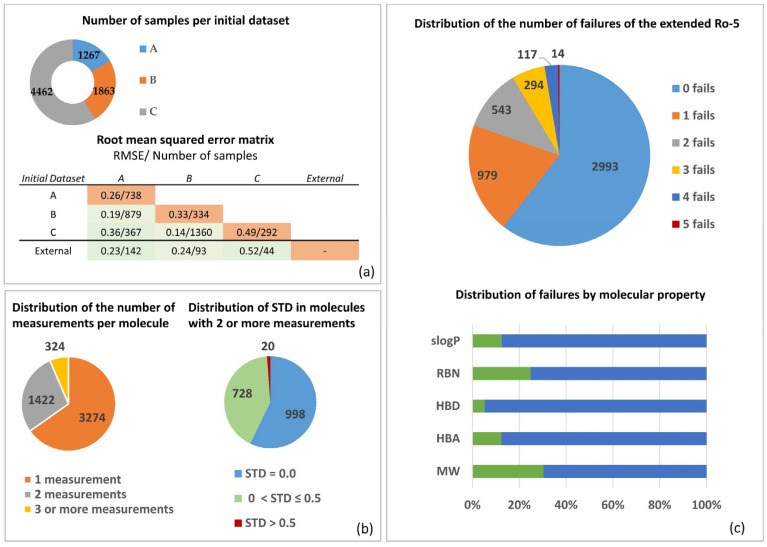
Exploration of MERGED Caco-2 permeability dataset: (**a**) Analysis of inter and intra overlapping of the individual curated Caco-2 permeability datasets with associated RMSE; (**b**) distribution with standard deviation (STD) of number of measures per molecule in the MERGED dataset; (**c**) distribution of extended rule-of-five number (Ro5) of failures in the MERGED data.

**Figure 2 pharmaceutics-14-01998-f002:**
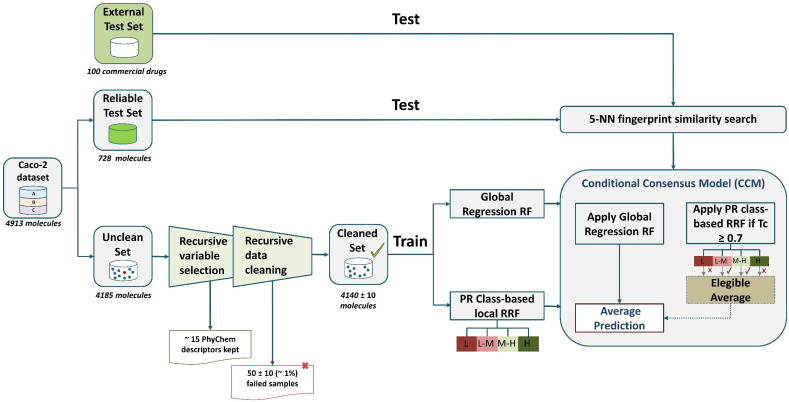
Overview of modeling and validation workflow. L: “low permeability” log *P_app_* < 6; L-M: “low–moderate” 6 ≤ log *P_app_* < 5; M-H: “moderate–high Permeability” 5 ≤ log *P_app_* < 4.7; H: “high permeability” 4.7 ≤ log *P_app_*. Tc: Tanimoto coefficient. RF: random forest. PR: permeability region.

**Figure 3 pharmaceutics-14-01998-f003:**
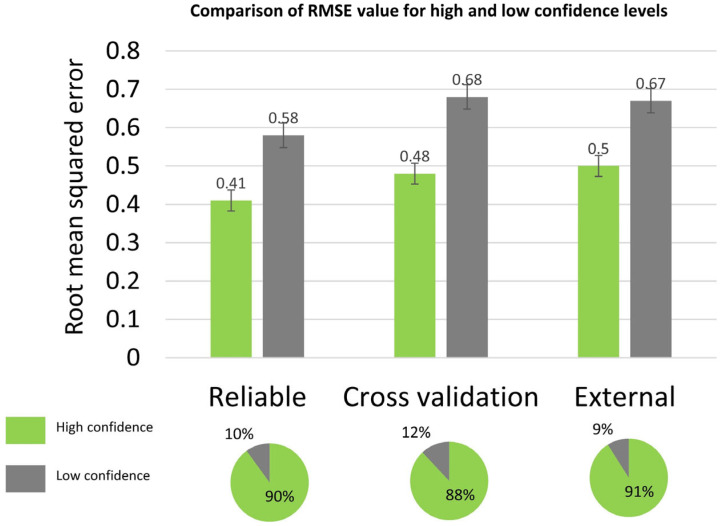
Comparison of the error metrics for high and low confidence levels. RMSE: root mean squared error. For each validation set, the pie chart shows the proportion of high and low confidence samples.

**Figure 4 pharmaceutics-14-01998-f004:**
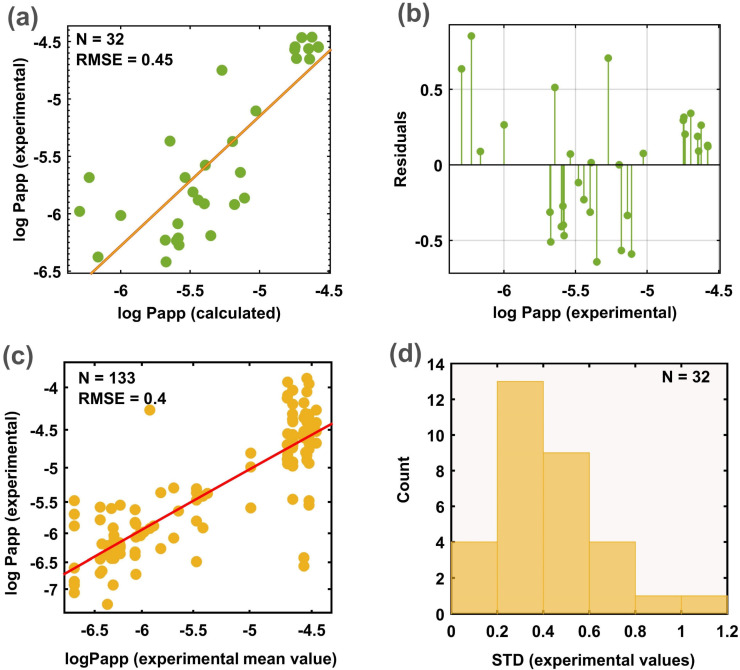
Overview of the model performance on the 32 reference drugs. (**a**) Regression plot for the 32 reference drugs; (**b**) residual plot; (**c**) correlation between the experimental mean value and all the experimental values reported for the same molecule; (**d**) standard deviation distribution of experimental values.

**Figure 5 pharmaceutics-14-01998-f005:**
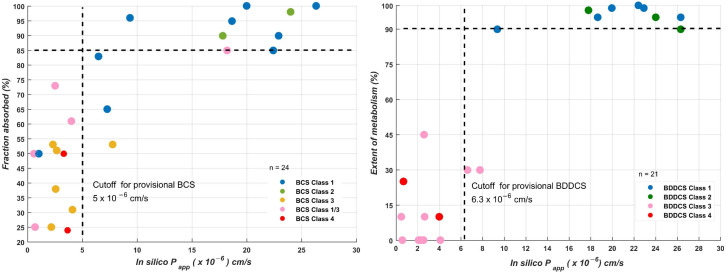
Relationship between in silico apparent permeability coefficients and fraction absorbed for the 32 reference drugs of the International Council Harmonization (ICH) list. Data points are classified according to BDDCS classes 1–4 or BCS 1–4.

**Table 1 pharmaceutics-14-01998-t001:** Validation metrics of conditional consensus model. Statistics: r^2^, r-squared of validation; RMSE, root mean squared error; MAE, mean absolute error; and % 0.5 log, percentage of molecules within the 0.5 log of prediction error.

Validation Set	N Mean (SD)	r^2^ _(validation)_ Mean (SD)	RMSE _(validation)_ Mean (SD)	MAE Mean (SD)	% 0.5 log Mean (SD)	N (Variables) Mean (SD)
Cross Validation	4166 (10)	0.54 (0.01)	0.51 (0.01)	0.39 (0.01)	70 (0.5)	15 (3)
Reliable Set	728	0.61 (0.01)	0.43 (0.01)	0.33 (0.01)	77 (0.6)	15 (3)
External Set	100	0.57 (0 03)	0.51 (0 03)	0.40 (0 01)	69 (3)	15 (3)

**Table 2 pharmaceutics-14-01998-t002:** Comparison between the current model and previous QSPR studies.

Study	Method	Data Size	Test Set Size	RMSE	NMAE *	r^2^	Units	Range
Wang et al. [24]	MESN	4464	1340	-	0.08	0.55	log *P_app_* (10^−6^ cm/s)	[−2.78; 2.48]
Wang et al. [22]	Boosting	1272	255	0.31	-	0.83	log *P_app_*	[−7.76; −3.51]
Wang and Shen [23]	HQPSO + dual RBF NN	1863	369	0.39	-	0.77	log *P_app_*	[−7.9; −3.72]
Sherer et al. [43]	RRF	15,791	-	0.2	-	0.52	log *P_app_*	−
Fredlund et al. [44]	PLS, RRF, SVM	2842	284	0.45	-	-	log *P_app_*	−
Current study	Recursive selection + RRF	5013 **	728	0.43	0.07	0.61	log *P_app_*	[−8.32; −3.51]
			100	0.51	0.08	0.57		

MESN: Multi-embedding-based synthetic network; NSGA: Non-dominated sorting genetic algorithm; HQPSO: Hybrid quantum particle swarm optimization; RBF: Radial basis function; NN: Neural networks; PLS: Partial least squared; SVM: Support vector machine; RRF: Regression random forest; NMAE: Normalized mean absolute error. * (Min, Max normalization). ** (This number includes the external validation set).

**Table 3 pharmaceutics-14-01998-t003:** Model outputs for 32 drugs included in the ICH list to validate and standardize Caco-2 permeability assay.

ICH Group Permeability	Drug	log *P_app_* obs.	log *P_app_* calc.	Drug Transport	Fabs * (%)	EoM (%)	BDDCS Class	BCS Class	BDDCS Predicted	BCS Predicted
High	Antipyrine	−4.55	−4.58	Passive diffusion	100	95	1	1	1, 2	1, 2
	Caffeine	−4.47	−4.7	Passive diffusion	100	99	1	1	1, 2	1, 2
	Carbamazepine	−4.57	−4.75	Passive diffusion	90	98	2	2	1, 2	1, 2
	Disopyramide	−5.37	−5.19	Passive diffusion	83		3	1	1, 2	3, 4
	Ketoprofen	−4.55	−4.58	Passive diffusion	100	90	2	1	1, 2	1, 2
	Metoprolol	−4.65	−4.73	Passive diffusion	95	95	1	1	1, 2	1, 2
	Minoxidil	−4.75	−5.27	Passive diffusion	95		1		3, 4	3, 4
	Naproxen	−4.46	−4.62	Passive diffusion	98	95	2	2	1, 2	1, 2
	Propranolol	−4.65	−4.64	Passive diffusion	90	99	1	1	1, 2	1, 2
	Theophylline	−5.11	−5.03	Passive diffusion	96	90	1	1	1, 2	3, 4
Moderate	Amiloride	−5.68	−6.23	Passive diffusion	50	0	3	1/3	3, 4	3, 4
	Atenolol	−6.27	−5.58	Paracellular	51	10	3	3	3, 4	3, 4
	Chlorpheniramine	−4.55	−4.74	Passive diffusion	85		1	1/3	1, 2	1, 2
	Enalapril	−5.64	−5.14	Passive diffusion	65		1	1	1, 2	3, 4
	Furosemide	−5.91	−5.4	Carrier-mediated	61	10	4	4	3, 4	3, 4
	Hydrochlorothiazide	−6.24	−5.6	Paracellular	73	0	3	3	3, 4	3, 4
	Metformin	−5.37	−5.64	Paracellular	53	0	3	3	3, 4	3, 4
	Ranitidine	−5.86	−5.11	Carrier-mediated	53	30	3	3	1, 2	3, 4
	Terbutaline	−5.92	−5.18	Paracellular	67	30	3		1, 2	3, 4
Low	Acyclovir	−6.38	−6.16	Paracellular Carrier-mediated	25	25	4	1/3	3, 4	3, 4
	Famotidine	−6.21	−5.59	Paracellular	38	45	3	3	3, 4	3, 4
	Chlorothiazide	−5.88	−5.44	Carrier-mediated	24		4	4	3, 4	3, 4
	Enalaprilat	−5.98	−6.3	Carrier-mediated	17	10	3		3, 4	3, 4
	Foscarnet	−6.19	−5.35	Paracellular	17		3		3, 4	3, 4
	Lisinopril	−6.42	−5.67	Carrier-mediated	25	0	3	3	3, 4	3, 4
	Mannitol	−6.09	−5.59	Carrier-mediated	20		3		3, 4	3, 4
	Nadolol	−5.58	−5.39	Paracellular	31	0	3	3	3, 4	3, 4
	Sulpiride	−6.23	−5.68	Carrier-mediated	36	0	3		3, 4	3, 4
Efflux	Digoxin	−6.02	−6	Carrier-mediated	50		3	1/3	3, 4	3, 4
	Paclitaxel	−5.81	−5.48	Carrier-mediated	50		2	4	3, 4	3, 4
	Quinidine	−4.56	−4.65	Carrier-mediated	85	100	1	1	1, 2	1, 2
	Vinblastine	−5.69	−5.54	Carrier-mediated	30		2		3, 4	3, 4

* Fabs: fraction absorbed. EoM: extent of metabolism. The log *P_app_* observed values refers to the mean values of the experimental values collected from the literature. Fraction absorbed and extent of metabolism data were taken from the literature.

**Table 4 pharmaceutics-14-01998-t004:** Comparison of the performance of this study and previously published models on the ICH drug list.

Study	Permeability/Metabolism Prediction Method for BCS/BDDCS	Number of Drugs from the ICH List	Sensitivity	Specificity
Golfar et al. (2019) [67]	clogP	29	0.89	0.81
Broccatelli et al. (2012) [65]	logD 7.5, logD 7, logD 9, log BB, CP,	25	1	0.64
Pham-The et al. (2013) [66]	logP, logD 7.5, TPSA, CACO2 (Volsurf+)	10	0.86	0.67
Kassim et al. (2004) [68]	logP, clogP	15	1	0.67
	In silico log *P_app_* for BCS	21	0.73	1
This study	In silico log *P_app_* for BDDCS	32	0.78	0.83

Sensitivity: TP/TP + FN; specificity: TN/TN + FP, where: TP: true positive; TN: true negative; FP: false positive; FN: false negative.

**Table 5 pharmaceutics-14-01998-t005:** Comparison of methods, variables, and model performance with previous in silico studies.

Study	Permeability Prediction Applied on:		Features	Methods	N (Test)	Statistics *
	BCS	BDDCS				
Broccatelli. (2012) [65]		x	17 features (Volsurf+ descriptors)	Bayes, SVM	379	Se: 0.7 Sp: 0.8
Golfar (2019) [67]		x	logP ACD	BLR	99	Acc: 0.86
Takagi [63]	x		logP, clogP	Comparison with cut-off value	29	Acc: 0.62–0.65
Pham-The 2013 [66]	x	x	logP, logD 7.5, TPSA, CACO2(Volsurf+)	LDA, BLR, QDA	675	Se: 0.73 Sp: 0.68
Khandelwal 2007 [69]		x	HBD, HBA, PSA, clogP, VOLSURF descriptors	RP, RF, SVM	56	Se: 0.86 Sp: 0.42
Dahan 2013 [62]	x		clogP, AlogP, KlogP	Comparison with cut-off value	29	Acc: 0.69–0.72
					14	Acc: 0.85–0.93
Newby 2014 [64]	x		Volsurf+ descriptors	DT	127	Se: 0.63 Sp: 0.62
This study	x	x	In silico log *P_app_* computed from 15 molecular descriptors using CCM	Comparison with cut-off value	204 (BDDCS)	Se: 0.8 Sp: 0.7 Acc: 0.77
					32 (BDDCS)	Se: 0.78 Sp: 0.83 Acc: 0.81
					22 (BCS)	Se: 0.67 Sp: 1 Acc: 0.82

* For studies where sensitivity (Se), specificity (Sp) or accuracy (Acc) values for the classification of permeability/extent of metabolism were not explicitly stated, statistics were calculated from the confusion matrix reported in the original articles. SVM, support vector machine; LDA, linear discriminant analysis; BLR, binary logistic regression; QDA, quadratic discriminant analysis; RP, recursive partitioning; RF, random forest; DT, decision tree.

## Data Availability

The training data and validation datasets are supplied in an XLSX file as part of the Appendix A. SMILES codes, log *P_app_* experimental value, log *P_app_* computational value, and RDKit descriptors are provided for each molecule. The open-source software KNIME version 4.4.2 (https://www.knime.com/download-previous-versions, accessed on 16 September 2022) was used for reading, analysis, modeling, and visualization of the data. Other extensions, such as “RDKit KNIME Integration” and “Indigo KNIME Integration” were downloaded directly from KNIME using the following update sites: https://update.knime.com/community-contributions/trusted/4.4 (accessed on 16 September 2022) and https://update.knime.com/community-contributions/4.4 (accessed on 16 September 2022).

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
