# Peer review of "Reliable Prediction of Caco-2 Permeability by Supervised Recursive Machine Learning Approaches"

_pharmaceutics, 2022, doi:10.3390/pharmaceutics14101998_

Round 1

Reviewer 1 Report

The authors dealt with the modelling of Caco-2 permeation of a wide range of molecules. The topic selection is important and may have high interest among the readers.

Nevertheless, there are some concerns which should be addressed before the acceptance of the paper.

The authors worked with a large and complex dataset. How were missing values treated? The 0 values in the data matrix are real zeroes, or may related to missing values too?

The stereoisomers may differ in their general effect, effectiveness, receptor affinity and even bioavailability their exclusion as duplicates may be a considerable error. An additional variable which identifies the enantiomers may be a better solution, or at least a small discussion about the deviation of the Papp values of the various stereoisomers, and how this deviation was treated during the duplicate elimination may be added.

Could it be that the different RMSE values of dupliactes in the various datasets may be related to the number of stereoisomers in the original datasets?

How were missing descriptors treated?

Minor comments

row 39 the dot is in wrong place.

row 232 the line breaking should be deleted

line 296 g/mo should be g/mol

Author Response

Answer to referees

Reviewer 1

The authors dealt with the modelling of Caco-2 permeation of a wide range of molecules. The topic selection is important and may have high interest among the readers.

Nevertheless, there are some concerns which should be addressed before the acceptance of the paper.

  1. The authors worked with a large and complex dataset. How were missing values treated? The 0 values in the data matrix are real zeroes, or may related to missing values too?

In the initial procedure of data curation, missing values of the target property were removed (listwise deletion). Values behind the cut-off -8.5 and -3 log units were removed too, due to their possible uncertainty. In the data matrix, zeroes correspond with real zeroes values in integer descriptors as Number of Aromatic Heterocycles, Number of Heteroatoms, ect.

  1. The stereoisomers may differ in their general effect, effectiveness, receptor affinity and even bioavailability their exclusion as duplicates may be a considerable error. An additional variable which identifies the enantiomers may be a better solution, or at least a small discussion about the deviation of the Papp values of the various stereoisomers, and how this deviation was treated during the duplicate elimination may be added.

No 3D descriptors were included in this study, but this issue was considered in the chemical and experimental data curation section. Molecules were considered duplicates if the standard deviation was less than 0.5, a rational limit considering the high variability of the target property. Of the 1836 molecules with more than one experimental value, only 9 pairs corresponded to stereoisomers. From this list, only one pair presented a significant difference between log Papp values, with a standard deviation of 0.79 (Pravastatin). This molecule was excluded from the modeling data set. The rest of the stereoisomers were in the 0.0-0.5 standard deviation range.

  1. Could it be that the different RMSE values of duplicates in the various datasets may be related to the number of stereoisomers in the original datasets?

Considering the small number of stereoisomers identified, we consider that the RMSE values reported in Figure 1 do not correspond to the presence of stereoisomers.

  1. How were missing descriptors treated?

In the case of molecular descriptors, the missing value treatment consisted of removing those columns with more than 10% of missing values. No missing values were reported for any RDKit descriptor.

  1. Minor comments

-row 39 the dot is in wrong place.

This was corrected in the text

-row 232 the line breaking should be deleted

This was fixed in the text

-line 296 g/mo should be g/mol

This was fixed in the text

Reviewer 2 Report

This article has developed a new methodology based on QSPR models to predict apparent permeability on Caco-2 cells during the early stages of drug discovery. Overall, the article is well-written and provides sufficient evidence to demonstrate the applicability of the methodology to serve as a useful tool for candidate's selection. 

Minor comments require further clarification by the authors: 

1. The Introduction helps to understand the need for this study and raises the current challenges in the field of research. However, I recommend adding the following reference (doi: 10.1016/j.vascn.2014.11.004) on line 65.

2. Is there any correlation between the number of experimental Papp values reported in the literature and the standard of deviation?

3. The methodology has demonstrated an excellent agreement between the observed and calculated log Papp values for highly permeable drugs (Table 3). However, the authors should clarify how many of the 32 drugs included in the ICH list were included in the reliable dataset or external dataset.

4. Figure 4a shows good agreement between experimental and predicted log Papp values <-5. However, larger discrepancies were observed for log Papp values >-5.5. What is the rationale behind this?

5. The authors stated on lines 479-480 the following sentence: The model allows to correctly identify 1005 of the molecules with moderate-low permeability and 91% of the molecules with high permeability. However, it is uncertain where this conclusion comes from. A clarification is needed. 

Author Response

Reviewer 2

This article has developed a new methodology based on QSPR models to predict apparent permeability on Caco-2 cells during the early stages of drug discovery. Overall, the article is well-written and provides sufficient evidence to demonstrate the applicability of the methodology to serve as a useful tool for candidate's selection. 

Minor comments require further clarification by the authors: 

  1. The Introduction helps to understand the need for this study and raises the current challenges in the field of research. However, I recommend adding the following reference (doi: 10.1016/j.vascn.2014.11.004) on line 65.

The reference was added to the manuscript

  1. Is there any correlation between the number of experimental Papp values reported in the literature and the standard of deviation?

In this regard, it is important to clarify that only 1836 molecules in our data set had more than 1 experimental measurement. For these molecules, a slight trend of decreasing standard deviation was observed with the increase of the number of experimental measurements.

  1. The methodology has demonstrated an excellent agreement between the observed and calculated log Papp values for highly permeable drugs (Table 3). However, the authors should clarify how many of the 32 drugs included in the ICH list were included in the reliable dataset or external dataset.

Of the 32 drugs in the ICH list, eighteen were included in the external set and eight in the reliable validation set. Neither of these two sets was used to predict the ICH drug list, and any overlap between the ICH list and the training set was removed prior to prediction.

  1. Figure 4a shows good agreement between experimental and predicted log Papp values <-5. However, larger discrepancies were observed for log Papp values >-5.5. What is the rationale behind this?

This fact can be attributed to several reasons. First, only 35% of the data covers the experimental range of -5.5 to -8.5. In addition, a slight tendency to increase experimental variability was observed for low permeability molecules. Most of the drugs with larger discrepancies are substrates of transporter proteins and this phenomenon cannot be completely covered by the physicochemical properties used by the model.

  1. The authors stated on lines 479-480 the following sentence: The model allows to correctly identify 1005 of the molecules with moderate-low permeability and 91% of the molecules with high permeability. However, it is uncertain where this conclusion comes from. A clarification is needed. 

In this sense, the sentence was rewritten as follows: “In correspondence with 10x10-6 cm/s (-5 log units) cutoff for high permeability, the model allows to correctly identify 100% of the molecules with moderate-low permeability and 89% of the molecules with high permeability.”